# Attention to Speech and Music in Young Children with Bilateral Cochlear Implants: A Pupillometry Study

**DOI:** 10.3390/jcm11061745

**Published:** 2022-03-21

**Authors:** Amanda Saksida, Sara Ghiselli, Lorenzo Picinali, Sara Pintonello, Saba Battelino, Eva Orzan

**Affiliations:** 1Institute for Maternal and Child Health—IRCCS “Burlo Garofolo”—Trieste, 34137 Trieste, Italy; sara.pintonello@burlo.trieste.it (S.P.); eva.orzan@burlo.trieste.it (E.O.); 2Ospedale Guglielmo da Saliceto, 29121 Piacenza, Italy; saraghiselli80@gmail.com; 3Dyson School of Design Engineering, Imperial College London, London SW7 2DB, UK; l.picinali@imperial.ac.uk; 4Faculty of Medicine, University of Ljubljana, University Medical Centre Ljubljana, SI-1000 Ljubljana, Slovenia; saba.battelino@kclj.si

**Keywords:** early bilateral cochlear implants, pupillometry, speech in noise, music in noise

## Abstract

Early bilateral cochlear implants (CIs) may enhance attention to speech, and reduce cognitive load in noisy environments. However, it is sometimes difficult to measure speech perception and listening effort, especially in very young children. Behavioral measures cannot always be obtained in young/uncooperative children, whereas objective measures are either difficult to assess or do not reliably correlate with behavioral measures. Recent studies have thus explored pupillometry as a possible objective measure. Here, pupillometry is introduced to assess attention to speech and music in noise in very young children with bilateral CIs (N = 14, age: 17–47 months), and in the age-matched group of normally-hearing (NH) children (N = 14, age: 22–48 months). The results show that the response to speech was affected by the presence of background noise only in children with CIs, but not NH children. Conversely, the presence of background noise altered pupil response to music only in in NH children. We conclude that whereas speech and music may receive comparable attention in comparable listening conditions, in young children with CIs, controlling for background noise affects attention to speech and speech processing more than in NH children. Potential implementations of the results for rehabilitation procedures are discussed.

## 1. Introduction

Selective attention to speech crucially determines the course of spoken language development. From birth on, typically-developing humans pay more attention to speech than to other auditory input [1,2,3], and selectively attend to relevant linguistic information useful for language development [4].

During the first months after birth, attention to speech is not related to the intelligibility of speech, but constitutes a prerequisite for language learning. In children with congenital hearing impairment (HI) who are born to hearing parents, this process is often disrupted due to the lack of auditory input, and, by consequence, the lack of linguistic input, prior to the implantation of cochlear implant (CI) [5]. Nowadays, most of the children with congenital HI receive (unilateral or bilateral) CIs before the first year of life. Nonetheless, they, at least initially, show a shorter attention span to speech than children with normal hearing (NH) [6]. This may play a role in their subsequent language development [7,8].

Later on in development, attention to speech is crucially related to the quality of listening conditions. In increased background noise, listeners tend to increase their selective attention to speech, which enables them to maintain speech intelligibility despite the degraded listening conditions [9]. Such an increase of selective attention elicits increased cognitive load in the context of listening, called listening effort, i.e., the allocation of cognitive resources to overcome obstacles or challenges to achieving listening-oriented goals [10]. When intelligibility becomes too low, or when listening is simply too effortful for any other reason, listeners often give up on the task, and their attention to speech decreases [11]. Noise may particularly disadvantage infants and young children in recognizing and learning from speech, especially when the background noise is also speech [12,13]. Because of its immediate detrimental effect on listening and overall quality of life [14,15], listening effort has received increased interest in the recent years. 

For listeners with CIs, speech perception (measured through speech intelligibility) is compromised at much lower levels of background noise compared to those with NH [16,17,18]. However, early implantation of bilateral CIs is thought to reduce these speech perception problems [19,20,21,22]. At least in some individuals, bilateral implantation yields reduced listening effort [23]. Furthermore, bilateral listening, either with bilateral cochlear implants or bimodal devices, improves music perception [24]. Nonetheless, speech perception in degraded listening conditions has not yet been measured in very young children with bilateral CIs, when language acquisition is in its peak. It is thus unknown how challenged their speech perception is compared to NH children, especially in complex auditory environments that occur in everyday life.

An increase of listening effort, caused by selective attention to speech and/or speech perception, can be measured with pupillometry. Namely, pupillary response induced by cognitive load is tightly linked to locus coeruleus (LC) neuron activation. The LC is a subcortical structure, and is the conductor of the noradrenergic system in the whole brain. The LC and norepinephric system are involved in different processes, including stress variations, memory retrieval, or selective attention [25,26]. The pupil diameter increases when it is difficult to focus attention on speech [27]. It appears to be a quadratic function of speech intelligibility, reaching peak values at medium levels of intelligibility during attentive listening [28], and decreasing again when intelligibility becomes too low [11,29]. However, listening effort is not to be confused with speech intelligibility score, as intelligibility can be compromised by other factors, such as mispronunciations or unknown words [30].

In adults with hearing loss, listening effort was measured in relation to speech intelligibility, speaking rate, semantic context, and auditory spectral degradation due to cochlear implant sound processing [28,31,32,33,34,35]. In pediatric populations with or without HI, there are only few studies related to listening effort using pupillometry. A study with NH school-age children has, notably, shown that an increased effort or increased attention elicits increased pupillary responses in children [36]. Another study investigated the effect of binaural fusion on listening effort in school-aged children with CIs, showing that lower binaural fusion caused slower response rates and an increase in pupil diameter, reflecting higher listening effort due to poorer speech perception [37]. The lack of studies with young children may be attributed to the absence of guidelines and references for this type of experiments, or to the fact that the potential pupil responses to cognitive processes may be difficult to disentangle from other physiological responses, such as responses to light and near-fixation [38]. Nonetheless, pupillometry does not require overt behavioural responses, and can, therefore, be administered also in non-communicative or non-collaborative subjects, such as infants. It is, therefore, a promising technique for the objective assessment of auditory perception [39], sound categorization [40], and language learning [41,42,43,44,45]. It has also been suggested that it may serve as a good method for assessing listening effort also in young children with congenital HI [38,46]. 

Because of a complete lack of studies on speech perception in noise in very young children measured with pupillometry, the aim of the present study was to explore pupillary responses during passive listening to speech and non-speech stimuli in young preschool children with early bilateral CI implantation, compared to the age-matched group of NH children. Children with CIs and children with NH were exposed to sequences of speech and music stimuli in varying listening conditions (with no, low, or high background noise levels). We hypothesized the relative increase of pupil response in conditions that elicit the highest selective attention and, consequently, the highest listening effort. The information about the relative attention allocated to specific auditory stimuli might contribute to fill in the existing gap in our knowledge about the impact of the listening effort on children with CIs, and further justify the necessity of controlling for noise levels when designing auditory environments for them, such as classrooms. 

## 2. Methods

### 2.1. Participant Sample

The study included a case group of children with CIs (CI group), and a control group of children with NH (NH group). The CI group consisted of 14 children (8 girls, 6 boys, mean age = 34.12 months, SD = 9.57, range: 16.73–47.03) with congenital HI and sequential bilateral CIs with normal cognitive abilities and motor skills (assessed as a part of clinical follow-up at 1 y after implantation with the Bayley Scales of Infant and Toddler Development, Third Edition (BSID-III)), and no visual deficits. The severe-to-profound non-syndromic sensorineural HI diagnosis had been made in the first months of life after a failed newborn hearing screening. The candidacy for CI surgery had been made after a period of traditional hearing aid amplification with limited or no benefit. All participants received the first CI before 24 months of age, with a variable interval between first and second surgery (Table 1). All children were regularly followed by the audiologist and speech therapists, and their mean-aided threshold was always fitted to ≤35 dB HL. Their language comprehension scores (assessed as a part of clinical follow-up at 1 y after implantation using a combination of a picture-naming task and Child Development Inventory scales [47,48], ranged from 5° to 90°). None of the children were bi-modally bilingual (raised in oral and sign language simultaneously). The final size of the case group was limited by the number of children who met the above criteria, and that we had access to at the medical institute where the study was conducted. Participants came from various regions in Italy, and had diverse socio-economic backgrounds. 

For the NH group, we tested 15 children (6 girls, 9 boys, mean age = 35.27 months (SD = 9.67, range: 22.47–48.73)) with normal hearing and with normal cognitive abilities and motor skills, and no visual deficits. Data from 1 child were not included in the final dataset because they failed to contribute trials with sufficient pupil data. Participants in the NH group came from the north-eastern regions in Italy, and had diverse socio-economic backgrounds. All parents were informed about the purpose of the study and the testing methods, and gave their written consent to participate before the testing began. Participants were treated in accordance with the Ethical Principles for Medical Research Involving Human Subjects (WMA Declaration of Helsinki). The study was approved by the regional medical Ethical Committee on 25/09/2018, ID: 2490; the testing was conducted in 2019 and 2020. 

### 2.2. Experimental Stimuli

The musical stimuli consisted of 5-s-long excerpts from the instrumental version of The Happy Song (http://www.imogenheap.com/thehappysong/, (accessed on 28 September 2018)). The Happy Song was composed for young children, and was designed specifically to elicit positive emotions [49,50]. The studio recording and the written consent to use the song in the study were obtained from the author of the song. The vocal part was excluded from the stimuli. The instruments included in the tracks were: synths, clarinet, guitar, mbira, and percussions. The excepts from the song were normalized for intensity to ensure that they were all at the same dBfs level (dBfs = dB full scale, the standard measure of the amplitude of digital signals). The individual left and right music channels were reproduced in different positions around the listener on the horizontal plane, in order to give the impression of the music coming from the surrounding environment.

Speech stimuli consisted of recordings of spoken rhymed verses in Italian. The verses were taken from an open-source website with lullabies and children’s songs (https://www.filastrocche.it/feste/filastrocche-per-bambini/, (accessed on 16 November 2018)). They were recorded by a native Italian female actress using child-directed speech in a studio, and normalized for intensity at the same dBfs level as music stimuli. The mono speech signal was played back through the individual loudspeaker (S0 at 0° elevation and 0° azimuth) during the test.

The noise was a multi-channel and multi-speaker speech noise recorded specifically for this project. It included at least 50 different talkers from multiple directions, and was then rendered through a loudspeaker array. Signal-to-noise ratios (SNR) were the following: no noise = only signal played, without the background noise; low noise = signal exceeding background noise for 10 dB (10-dB SNR); high noise = signal at the same output level as the background noise (0-dB SNR)). The decision to use these values was based on studies where ∼0-dB SNR resulted in 50% speech intelligibility for HI, and 80% for NH adults, with intelligibility exceeding 95% intelligibility for ∼7-dB SNR in both groups [34,35]. Although to our knowledge, no such study exists for children with HI, toddlers with normal hearing were shown to exploit the speech environment at 10-dB SNR, but not at 5-dB SNR [12]. 

The visual stimuli consisted of one 5-s-long excerpt from a silenced version of an animated film (https://koyaa.net/en/, (accessed on 13 March 2019)) that was presented in all trials, and 27 still frames from the same film presented during the inter-stimulus-intervals. The HD copy and the written consent to use the film in the study were obtained from the author of the animated film. The content of the visual input was not meaningfully related to the speech input. The choice for an animated colorful visual stimulus was based on several existing studies [39,40,44]. Possible interferences due to the properties of the stimulus (changes in luminance because of movements and changes in colours) were controlled by the fact that the excerpt from the film was always the same. Furthermore, general physical conditions (light conditions in the room and the setup) matched across all trials. Auditory and visual stimuli used in this study are available publicly at https://osf.io/7smcx/?view_only=f42352907af84ec5afdd73aea5cce239. 

### 2.3. Procedure

Each child was tested separately in a quiet experimental room. The total duration of the testing session was ∼5 min. During the test, children passively listened to the excerpts of music or speech, with or without background noise, while watching the visual stimulus on the eye-tracking screen. No other task was required from children. There were 9 experimental conditions: each type of signal (speech, music, or silence) was presented in various types of background noise (high, low, no noise). Each combination of signal and noise levels was presented three times, resulting in total 27 trials. The trial structure is presented in Figure 1A. During the presentation of different trials, children watched the 5-s-long excerpt from the animated film, with various still frames from the same film presented during the inter-stimulus-intervals.

### 2.4. Apparatus

The toddlers’ gaze and pupil size were recorded with a TOBII T120 eye-tracker at 60 Hz. Due to low head movement errors, TOBII T120 allows for extensive freedom of head movement without chin rests or movement-restricting headgear. The eye-tracker was integrated into a 17-inch TFT screen. Infants were seated on their parent’s lap at about 60 cm distance from the screen. Parents wore blocked glasses or kept their eyes shut to avoid the eye-tracker collecting their gaze, and to prevent them from influencing infants’ looking behavior. Visual stimuli, and the commands for the eye-tracker and auditory stimuli were presented through the Psyscope X software (http://psy.ck.sissa.it, (accessed on 5 October 2018)).

The 3D audio simulation was create” usi’g a custom MaxMSP patch implementing 4th Order Ambisonics encoding and decoding [51,52]. The computer was connected to an RME MADIFace audio interface and to a Sonible d:24mio amplifier, which delivered the analogue signals to a 24-channel spherical loudspeaker array. The loudspeakers on the system were distributed on the surface of a pseudo-sphere, at a distance between 1 and 1.2 m from the listeners’ position. Their distribution was uniform and denser on the horizontal plane, where 12 loudspeakers were located. The other 12 loudspeakers were located on two rings (8 and 4 loudspeakers, respectively) above the listener. Each loudspeaker was individually calibrated, ensuring that the level, frequency response, and time of arrival were consistent across the various elements of the array [51,52]. The system was calibrated so that each type of signal generated 60 dB SPL at the listener position. Final output levels of various combinations of the signal and noise resulted in a relative increase of overall intensity. The final sound pressure levels were measured at the center of the experimental room, and are reported in Table 2. The experimental setup is presented in Figure 1B.

### 2.5. Data Analysis and Data Availability

Pupil size output in the TOBII T120 eye-tracker is already corrected for the gaze and head position (https://www.slideshare.net/AcuityETS/tobii-t60-t120-user-manual, (accessed on 15 October 2019)). The prepossessing thus included the removal of artifacts or outliers (data lower than the 5th or higher than the 95th percentile), the selection of valid data-points (measured when gaze was fixated to the screen), and the exclusion of trials in which more than 25% of the pupil data were missing during the test phase [40,53]. This way, 14.5% of recorded trials in both groups were removed. The missing data in the included trials were linearly interpolated and then extrapolated using mean values of the trial. Finally, pupil data were baseline-corrected using subtractive baseline correction based on the average of the first 200ms at the beginning of each test phase. All analyses and visualizations were programmed in R [54,55,56]. The anonymized dataset including raw pupil data and external variables is publicly available, along with the analysis script, at https://osf.io/7smcx/?view_only=f42352907af84ec5afdd73aea5cce239.

## 3. Results

The mean differences in pupil responses based on the condition-based manipulations were assessed with a permutation linear mixed effect model (GLMM; [56]), in which the dependent variable was the average pupil response for each trial in each participant, and the fixed factors were noise (no noise, low noise, and high noise), signal (music, speech), and group (HN, CI), and their interactions, with random intercepts for subjects, trials, and sound output levels. We used the buildmer function in R to test which was the maximal feasible model given the data. If included, the interaction terms could tell us about the possible effects of noise to different signals in the two groups. However, these interactions could not be included in the final model without resulting in either an over-fit or the failure of the model to converge. The exclusion of the interaction from the final model also showed that the data were not sufficient to split them into two groups. The maximal feasible model given the existing data thus resulted in the exclusion of the interactions between the fixed factors, and the exclusion of sound output levels from the random structure. The corresponding permutation p-values for the maximal model were obtained through the general permutation test for mixed-effects models [57]. The advantage of these permutation p-values is that multiple models can be run without requiring any Bonferroni correction. Type III Analysis of Variance of the model with Satterthwaite’s method showed a significant effect of noise (F(2) = 4.29, *p* = 0.014), in that high background noise elicited a higher response than no noise (estimate = 0.05, sd = 0.02, *t* = 2.16, *p* = 0.003); a significant difference between the two groups (F(1) = 5.46, *p* = 0.019), in that the HN group had overall larger pupil values across conditions (estimate = 0.05, sd = 0.02, *t* = 2.16, *p* = 0.003); and significantly different responses to signals (F(2) = 3.86, *p* = 0.021), with both music and speech eliciting a larger pupil size compared to silent trials (music: estimate = 0.06, sd = 0.02, *t* = 2.734, *p* < 0.001; speech: estimate = 0.04, sd = 0.02, *t* = 1.826, *p* = 0.013). The variance among participants did not exceed the magnitude of the fixed effects. Visual inspection of residual plots (Appendix A) showed the predicted heteroscedasticity and low autocorrelation, indicating that the model is valid in predicting the effects. Mean responses to different signals and different noise levels are presented in Figure 2. Further details of the analysis are presented in Appendix A.

The presented analysis revealed the overall effects of group, noise, and signal, but not whether the background noise affected responses to music and speech, and how. Given that the existing data did not permit to test these interactions with averaged pupil values, and given that the responses may have differed only in some restricted time-windows, we conducted further analyses using all time-samples in the data (long data). To assess the time course of the differences between conditions, two permutation-based time-course analyses [58] were run for each group using the long data: one to assess time-windows in which the presence of background noise elicited differences in pupil size during speech trials, and one to assess comparable differences in pupil size during music trials. The permutation tests were based on the restricted likelihood ratio test statistic [59]. The time-windows in which the responses to the tested conditions were different were estimated by computing the cluster-based statistic using the using permuted likelihood ratio tests [57,58]. Although this could already comprise the entirety of the analysis, we used the time samples selected with the cluster-mass test as an input for further analyses (an approach common in various fields that use time-course data). For each analysis, the values in the time-window from the first until the last significant time-sample were averaged, and taken as an input for a permutation linear mixed-effects model that allowed us to obtain robust *p*-values. 

In children with CIs, pupil responses to speech in noise were lower than responses to speech in no noise in several (but not all, as seen from Appendix A) data samples during the time-window between 0.20 and 3.45 s. The permutation linear mixed effect model run over the average values in this time-window revealed marginally significant differences both for the low (estimate = −0.08, sd = 0.06, *t* = −1.29, *p* = 0.07) and for the high noise levels (estimate = −0.08, sd = 0.06, *t* = −1.32, *p* = 0.06). Conversely, we found no significant time-window in the NH group. The time-course of the responses to speech in noise, along with the time-windows of significant differences computed with the cluster-based statistic, are presented in Figure 3A. The outputs of the permutation-based analyses are presented in Appendix A. 

For responses to music in noise, the permutation analyses revealed a different pattern. In the CI group, the response to music in noise elicited a significant increase in pupil size in two narrow time-windows between 0.48 and 2.18 s, but the permutation linear mixed effect model run over the average values in this time-window revealed no significant differences between the noise conditions. Conversely, children in NH group responded with a significantly increased pupil size throughout the trial when the background noise was high (estimate = 0.24, sd = 0.06, *t* = 4.22, *p* < 0.001), but not significantly when the background noise was low. The time-course of the responses to music in noise with time-windows of significant differences computed with the cluster-based statistic are presented in Figure 3B. The outputs of the permutation-based analyses are presented in Appendix A. 

These results would be all the more interesting if pupil data could be compared directly to the comprehension of presented speech stimuli. Pupil size difference between noisy and non-noisy speech trials could, for example, (possibly negatively) correlate to the speech comprehension measure [28]. However, at least some children included in the study were not yet able to speak, and hence, could not be tested for the comprehension of the presented stimuli without an additional independent experimental design. Therefore, the present study only involved pupillometry and the indirect measures of cognitive and language comprehension obtained during regular clinical follow-up exams through parental questionnaires (CDI scale) and a picture-naming task. These indirect measures were not expected to have a strong correlation to pupil data, given the large difference between the stimuli in the passive listening task during pupillometry (consisting of listening to spoken rhymed verses) and the stimuli used in clinical evaluations of language development. Nonetheless, as an exploratory analysis, we ran the multiple correlations between these indirect measures and the average difference between pupil size during speech without noise, and speech with background noise, for each participant in the CI group (spin.diff variable). The results showed no correlation between spin.diff and language comprehension scores reported in Table 1. They, conversely, showed a significant correlation between IQ-motor scores and language scores (R(11) = 0.85, *p* = 0.001), which has been attested previously in literature [60], and a marginally significant correlation between the spin.diff variable and the time since the implantation of the first CI (Ci1time) (R = 0.53, *p* = 0.09), indicating the benefit of the time spent using the CIs [7]. 

## 4. Discussion

The analyses of the pupil data in this study revealed several interesting points. Importantly, all three main factors in the study elicited significant differences in pupil size. The overall bigger pupil size in children with NH was somewhat unexpected. It was not related to direct auditory manipulations during the experiment, and may, therefore, have an external cause. It is possible that the difference was elicited by the experimental setting: children with CIs were familiar with the locus of the experimental setup because they were regularly followed at the medical institute, whereas NH children were only there for the first or second time. The overall excitement due to the novelty of the environment may have contributed to the overall increase of pupil size in these children. Another two effects were directly linked to the auditory changes during the experiment. The fact that the presence of a signal (speech or music) elicited a larger pupil size compared to trials without any signal, and, therefore, enhanced selective attention, is in line with the recent preliminary data on pupillometry as an index of selective auditory attention or the effort of auditory attention switching [30,61,62]. Similarly, the presence of (high) background noise elicited an overall larger pupil size compared to no-noise trials. 

However, as observable from Figure 3, and confirmed during the permutation-based time-course analysis, high background noise elicited an overall larger pupil size only in the NH group, and the effect was largely driven by their increased response to music in high noise. In children with NH, listening to music in background noise may have elicited a relative increase of attention because it was composed specifically to attract young children [49,50], and it is, therefore, possible that pupil size increased when children directed additional effort into hearing the tune in high background noise. Conversely, no such effect was observed in children with CIs. The results are informative about the overall difference in selective attention allocated to speech and music in noisy conditions, and the overall differences between the CI and NH groups. However, the nature of the stimuli used in this study does not allow us to extract more information about music perception in noise in young children with CIs. Namely, the presented music excerpts were instrumentally and rhythmically complex sounds, and only a more comprehensive study that would control for these characteristics could enable a better understanding of the attentional mechanisms for music perception in young children with CIs.

The permutation-based time-course analysis also revealed that the general increase in response to high background noise was reversed in the CI group when they listened to speech. Children with CIs responded to speech in the absence of background noise with a relative increase in pupil size compared to speech in background noise, whereas such a response was missing in children with NH (Figure 3A). It is possible that the difference between the background noise did not affect listening to speech in NH children because the presented levels of background noise did not compromise intelligibility, whereas for children with Cis, intelligibility, and, consequently, attention to speech, significantly decreased when noise was added. We used babble (50-talker) noise as a background noise to create an ecologically valid replication of the everyday environment. In NH adults, such a modulated masker noise masks “less” compared to continuous maskers, such as pink or white noise [63,64], which strengthened our prediction that speech will remain intelligible in noise at least for normally-hearing children. For adult CI users, the effects of such multi-talker maskers have only been studied in the restricted cases of 1- and 4-talker maskers; however, with the results in line with previous studies [35]. Additionally, we also recreated a virtual 3D acoustic environment to make the task more ecologically valid. Such virtual acoustic realities have been recently shown to increase the difficulty of the listening task compared to lab-based assessments due to the added complexity, especially in patients with HI [65,66]. In the presence of a virtual 3D acoustic environment, it is possible that the intelligibility decreased excessively in both high and low background noise, and that young children with CIs subsequently found the task of listening to speech too demanding when background noise was present. 

Due to the participants’ young age, the present study could not include a subjective measure of speech perception for the presented speech stimuli, which is otherwise a standard procedure in studies with adults [28,33,34]. Such a procedure would be possible in older children and adults; however, the present study included age-specific stimuli (spoken rhymed verses, and the excerpt from an animated film suitable for children under 5 years of age). More importantly, it would be extremely difficult to obtain a sample of older children or adults with anamnestic characteristics comparable to our CI group sample. Such an extension of the study was, therefore, beyond the reach of the present study. Because of this, we lack certainty that the observed effects in pupil size indeed reflect differences in speech perception among the CI and NH group. As predicted given the large difference between the stimuli used for pupillometry and the stimuli for clinical evaluations of speech comprehension, there was no clear correlation between the clinical language comprehension scores and the differences in pupil size during different speech conditions. The results of this study may nonetheless offer an additional insight into the vulnerability of speech perception in young children with CIs whose every day is comprised of listening in a complex three-dimensional noisy acoustic environment. 

Finally, given that this is one of the first studies that observes pupil responses to various auditory stimuli in young children with CIs, we would like to point out the methodological note of the study. The comparison of different conditions within this study was possible because visual stimuli remained unchanged across all trials, eliciting a regular phasic change visible in all trials [67] (see Figure 3 for comparison). The differences could, therefore, only be attributed to changes in auditory input. Importantly, although each child contributed only up to three trials per condition in each testing session, the proposed analysis shows condition-based differences in the pupillary responses of young children with CIs, a group of highly variable and less communicative subjects. This is important because the majority of the existing speech perception tests for children with HI require an overt behavioral response, and are, therefore, hard to administer in very young children. The lack of precise speech perception measures can compromise a range of rehabilitation processes that are crucial for positive language outcomes, for example, CI fitting and speech therapy. In the near future, pupillometry may, therefore, prove as an objective measure that can be used with infants and young children. As such, it could prove efficient for exploratory studies on speech perception and rehabilitation procedures involving infants and young children with CIs.

## 5. Conclusions

To our knowledge, this is the first study that objectively tests selective auditory attention to speech and non-speech stimuli in background noise in very young children with bilateral CIs using pupillometry. It shows that young children with bilateral CIs attend to speech and music, but that the attention to speech may improve in non-degraded listening conditions (conditions without background noise). These results support the conclusions from previous studies about the increased sensitivity to noise in pediatric populations with CIs [16,18]. A good signal-to-noise ratio could, therefore, further enhance attention to speech in young children with CIs, which is a prerequisite for good language outcomes. Though a more comprehensive overview is required for a better understanding of attention mechanisms for music perception in young children, the observed pupil responses also point to the categorical differences in processing speech and music. Combined, these results confirm the predictions that pupillometry may be used in exploratory studies, and, potentially, also in rehabilitation procedures involving very young children with CIs. 

## Figures and Tables

**Figure 1 jcm-11-01745-f001:**
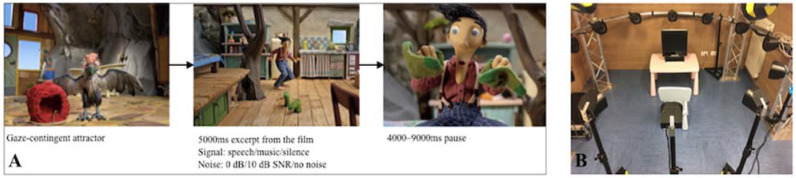
(**A**) The schematic representation of the trial structure. (**B**) A photo of the experimental setup. Older children were seated on the chair in front of the screen; younger children were seated in their parents’ lap.

**Figure 2 jcm-11-01745-f002:**
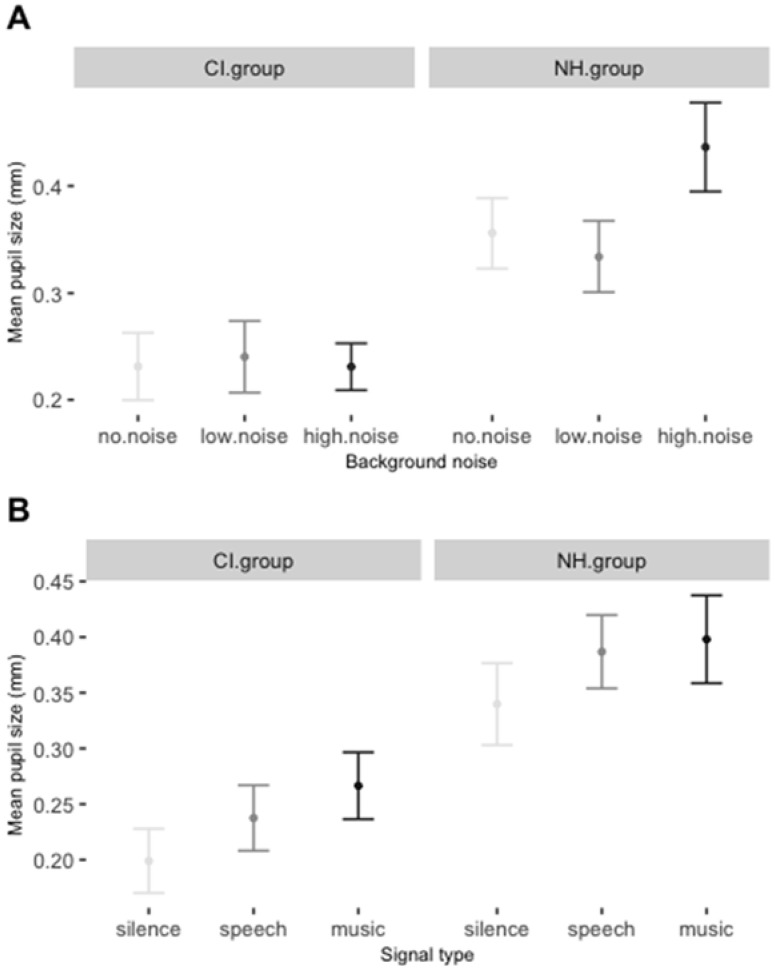
(**A**) Mean pupil responses per group with noise as the averaging factor. (**B**) Mean pupil responses per group with signal as the averaging factor. Error bars represent standard error rates.

**Figure 3 jcm-11-01745-f003:**
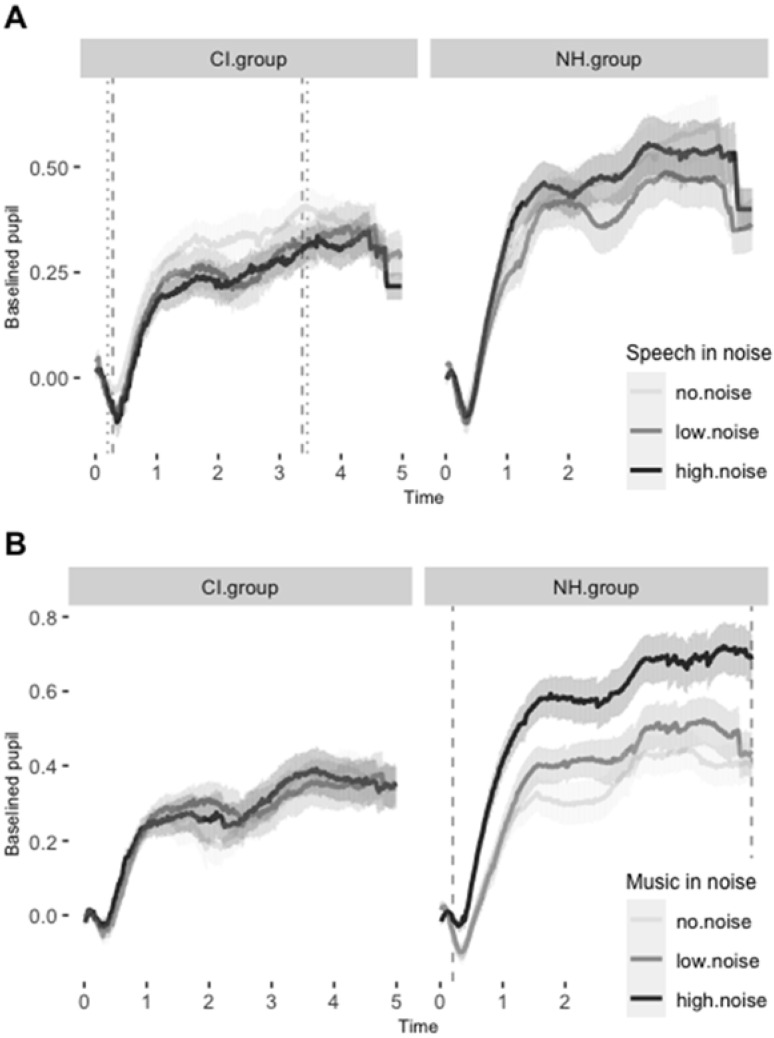
(**A**) The averaged time course of the pupillary responses to speech in various noise conditions in both groups. The dashed vertical lines represent the (cluster-based) estimate of the time-window of marginally significant differences between No and High noise conditions. The dotted vertical lines represent the time-window of marginally significant differences between No and Low noise conditions. (**B**) The averaged time course of the pupillary responses to music in various noise conditions in both groups. The dashed vertical lines represent the (cluster-based) estimate of the time-window of significant differences between No and High noise conditions.

**Table 1 jcm-11-01745-t001:** Participant data for the CI group.

Subject Code º	Sex	PTA * with CIs	IQ Verbal	IQ Non-verbal	IQ Motor	Age at Testing (Months)	Age at CI1 (Months)	Age at CI2 (Months)	Time from CI1 (Months)	Language Score **	Bilingual in Two Oral Languages (0 = no, 1 = yes)
**CI1**	F	27	86	110	97	47.03	29.5	40.3	17.53	50	0
**CI2**	F	28	77	100	94	34.83	10.43	15.27	24.4	10	0
**CI3**	M	26	95	122		45.33	20.9	28.9	24.43	50	0
**CI4**	M	25	89	105	97	40.77	11.93	15.77	28.83	50	0
**CI5**	F	27	86	95	91	16.73	11.83	14.8	4.9	10	0
**CI6**	F	34	59	85	85	30	26.03	29.13	3.97	5	0
**CI7**	M	28	69	100	82	18.57	12.03	12.17	6.53	5	0
**CI8**	F	31	86	97		31.87	11.07	15.5	20.8	10	1
**CI9**	M	31	94	105	97	32	10.73	24.47	21.27	50	0
**CI10**	M	32				39.7	18.97	33.67	20.73	5	1
**CI11**	F	31	71	90	100	36.53	25.17	20.67	15.87	90	1
**CI12**	F	31	97	110	107	32.2	10.7	12.77	21.5	90	1
**CI13**	M	34	65	96	94	46.93	21.73	29.37	25.2	5	0
**CI14**	F	31	83	110	97	25.23	13.77	19.63	11.47	50	0
MEAN		29.71	81.31	101.92	94.64	34.12	16.77	22.32	17.67	34.29	
SD		2.87	12.12	9.84	6.86	9.57	6.74	8.75	8.02	31.19	

º Participants were assigned ordinal numbers according to the order of testing; * Pure Tone Average = average air-tonal threshold at the 500, 1000, 2000 and 4000 Hz. PTA is expressed in dB HL; ** Language comprehension measures (picture naming task and Child Development Inventory scales) at 1 year post CI implantation evaluated in percentile scores.

**Table 2 jcm-11-01745-t002:** The final intensity levels in the testing room.

Signal	Noise	Output Pressure (dB SPL)
silence	no noise	48
	low noise	56
	high noise	66
speech	no noise	63
	low noise	64
	high noise	68
music	no noise	62
	low noise	65
	high noise	69

## Data Availability

The dataset is presented for publication for the first time, and we are not aware of any other comparable or duplicate published study. The anonymized dataset including raw pupil data and external variables is publicly available, along with the analysis script and stimuli used in the study, at https://osf.io/7smcx/?view_only=f42352907af84ec5afdd73aea5cce239.

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
