# Peer review of "Attention to Speech and Music in Young Children with Bilateral Cochlear Implants: A Pupillometry Study"

_jcm, 2022, doi:10.3390/jcm11061745_

Round 1

Reviewer 1 Report

This is indeed a novel way to describe the objective measurement of speech perception. Visual pathway plays an important role in learning and attention and this potential has been very well harnessed by the authors. The experimental design of this study is well thought out but it is difficult to reproduce these experimental settings in usual audiology and rehabilitative settings. Further research would be needed to make it simpler and easier to measure methods. 

The idea of using pupillometry as a measure of speech perception in noise and quiet surroundings in CI patients is unique and hence should be published but whether it can be made into a real-life easily reproducible model of measurement of speech perception would have to be seen in near future.

Author Response

We thank Reviewer 1 for the positive review of the manuscript. 

We agree that it is difficult to reproduce the experimental settings from the study in usual audiology and rehabilitative settings, and that further research is needed to transform it into a simpler and easier-to-measure method.

Reviewer 2 Report

This study by Saksida et al. used pupillometry to objectively test selective auditory attention to speech and non-speech stimuli in background noise in very young children with bilateral CIs. It shows that young children with bilateral CIs attend to speech and music but that the attention to speech may improve in ideal conditions. Their results confirm the conclusions from previous studies about the increased sensitivity to noise in the pediatric population with CIs⁠. Therefore, a good signal-to-noise ratio could further enhance attention to speech in young children with CIs, which is a prerequisite for good language outcomes. This study suggests that pupillometry may be used in exploratory studies and potentially also in rehabilitation procedures involving very young children with CIs. The data are interesting, and the experiments seem well performed. Although the group size is small, the results also have the potential for future clinical usage.

  1. In line 14, the age range of the control group should be added.
  2. Many sentences are hard to follow. For example, in line 55, It is… .
  3. The quality of the table should be improved. For example, in table 1, age at CI2 is confusing. The (y) is not for years. The whole table should be reorganized.
  4. In line 119, the author mentioned that none of the children was bi-modally bilingual. I thought all CIs in this study were bilateral for the table.
  5. In line 127, Additional 1 child…., what is the exact meaning of this sentence?
  6. In line 143, what is dBfs?

Author Response

We thank the Reviewer 2 for the comments, which were addressed in the revised version of the manuscript. The responses can be found below each point.

  1. In line 14, the age range of the control group should be added.

Thank you for noticing; the age range is now added.

  1. Many sentences are hard to follow. For example, in line 55, It is… .

Thank you for pointing out. We have now transformed the sentence as following:

“Nonetheless, speech perception in degraded listening conditions has not yet been measured in very young children with bilateral CIs, when language acquisition is in its peak. It is thus unknown how challenged their speech perception is compared to NH children, especially in complex auditory environments that occur in the everyday life.”

  1. The quality of the table should be improved. For example, in table 1, age at CI2 is confusing. The (y) is not for years. The whole table should be reorganized.

All ages are transformed into months now, in the table and in the text.

  1. In line 119, the author mentioned that none of the children was bi-modally bilingual. I thought all CIs in this study were bilateral for the table.

None of the children was bi-modally bilingual (raised in oral and sign language simultaneously).

  1. In line 127, Additional 1 child…., what is the exact meaning of this sentence?

The sentences are now slightly transformed to enhance clarity: “For the NH group, we tested 15 children (6 girls, 9 boys, mean age = 35.27 months (SD = 9.67, range: 22.47-48.73)) with normal hearing and with normal cognitive abilities and motor skills, and no visual deficits. Data from 1 child were not included in the final dataset because they failed to contribute trials with sufficient pupil data.”

  1. In line 143, what is dBfs?

Explanation added in the brackets: “(dBfs = dB full scale, the standard measure of the amplitude of digital signals)”

Reviewer 3 Report

This pupillometry study tested response to speech and music in varying noise conditions in CI toddlers compared to NH controls. Auditory stimuli were presented in 3 noise conditions; no noise, low noise, and high noise, and the pupil size was compared across conditions and groups. Regarding the difficulty of obtaining an objective measure for listening in young CI recipients, the method has the potential for further development. However, I have some concerns about the manuscript.

  1. In analyzing mean pupil sizes (Figure 2), the authors reported a significant effect of noise and signal. However, as shown in figure 2A, the effects would be applied differently between groups. The noise effect might not be significant in the CI group, but this was not analyzed in the manuscript.
  2. line 279-281: The results of marginal significance should be reported with great care. In figure 3, those results were marked as significant ones. Additionally, in line 291, the low noise in music was said to be not significant, but it was marked to be significant in figure 3B right panel.
  3. The method looks promising regarding the difficulty of obtaining an objective measure for listening in young CI recipients. If the pupil size were related to any measure of speech comprehension, the result would be valuable. With the wide variability of language comprehension scores presented in Table 1, this type of analysis might be possible, but it was not shown in the manuscript.
  4. Understandably, the NH children show an overall higher response than CI children to auditory stimuli. For CI users, listening to speech in noise is too difficult to attend to the stimuli. Listening to music is not entirely different from listening to noise. And the result of the study shows the stimuli used in this study were not suitable for CI children to draw their attention.

Minor concerns:

  • Table 1. PTA -> CI threshold
  • Table 1, Age at CI2(y) -> (m) or convert the data as of year format.
  • I do not think supplementary materials 1 and 2 are necessary.

Author Response

We thank the Reviewer 3 for the valuable observations and we believe addressing the issues pointed out improved the article substantially. The responses can be found below each point.

  1. In analyzing mean pupil sizes (Figure 2), the authors reported a significant effect of noise and signal. However, as shown in figure 2A, the effects would be applied differently between groups. The noise effect might not be significant in the CI group, but this was not analyzed in the manuscript.

We used buildmer function in R to test which is the maximal feasible model given the data. The maximal proposed model included also interactions between group, noise, and signal. If included, they could tell us about possible effects of noise to different signals in the two groups. However, these interactions could not be included in the final model without resulting in either an over-fit or the failure of the model to converge.

The exclusion of the interaction from the final model also showed that the data were not sufficient to split them into two groups without including the time course analysis. The above explanation is now included in the results section for further clarity.

  1. line 279-281: The results of marginal significance should be reported with great care. In figure 3, those results were marked as significant ones. Additionally, in line 291, the low noise in music was said to be not significant, but it was marked to be significant in figure 3B right panel.

Thank you for pointing out the inconsistency. In fact, the figures in Fig 3 were done before running the permutation linear mixed effect model (LMER) over the average values in the time windows selected with the time-course cluster-based permutation analysis, and the lines represented the time windows selected with the latter. We have now corrected the figures and included only the time windows that were (marginally) significant after the permutation LMERs.

We included marginally significant analyses because in the final permutation LMERs, the averaged time-windows included also several non-significant data samples.

  1. The method looks promising regarding the difficulty of obtaining an objective measure for listening in young CI recipients. If the pupil size were related to any measure of speech comprehension, the result would be valuable. With the wide variability of language comprehension scores presented in Table 1, this type of analysis might be possible, but it was not shown in the manuscript.

We agree with your observation that the results would be all the more valuable if related directly to a speech comprehension score related directly to the presented speech stimuli. We have now added the following paragraphs to the results and discussion section, which we believe addresses your concern and the concern of future readers of the article:

“These results would be all the more interesting if pupil data could be compared  directly to the comprehension of presented speech stimuli. Pupil size difference between noisy and non-noisy speech trials could for example (possibly negatively) correlate to the speech comprehension measure [28]. However, at least some children included in the study were not yet able to speak and hence could not be tested for the comprehension of the presented stimuli without an additional independent experimental design. Therefore, the present study only involved pupillometry and the indirect measures of cognitive and language comprehension obtained during regular clinical follow-up exams through parental questionnaires (CDI scale) and picture naming task. These indirect measures were not expected to have a strong correlation to pupil data, given the large difference between the stimuli in the passive listening task during pupillometry (consisting of listening to spoken rhymed verses) and the stimuli   used in clinical evaluations of language development. Nonetheless, as an exploratory analysis we run the multiple correlations between these indirect measures and the average difference between pupil size during speech without noise and speech with background noise for each participant in the CI group (Spin.diff variable). The results showed no correlation between Spin.diff and language comprehension scores reported in Table 1. They, conversely, showed a significant correlation between IQ-motor scores and language scores (R(11) = 0.85, p = 0.001), which has been attested previously in literature [60], and the marginally significant correlation between the spin.diff variable and the time since the implantation of the first CI (Ci1time) (R = 0.53, p = 0.09), indicating the benefit of the time spent using the CIs [7]. 

“Due to the participants’ young age, the present study could not include a subjective measure of speech perception for the presented speech stimuli, which is otherwise a standard procedure in studies with adults [28],[33],[34]. Such procedure would be possible in older children and adults; however, the present study included age-specific stimuli (spoken rhymed verses and the excerpt from an animated film suitable for children under 5 years of age). More importantly, it would be extremely difficult to obtain a sample of older children or adults with anamnestic characteristics comparable to our CI group sample. Such extension of the study was therefore beyond the reach of the present study. Because of this, we lack certainty that the observed effects in pupil size indeed reflect differences in speech perception among the CI and NH group. And as predicted given the large difference between the stimuli used for pupillometry and the stimuli for clinical evaluations of speech comprehension, there was no clear correlation between the language comprehension scores and the differences in pupil size different speech conditions. The results of this study may nonetheless offer an additional insight into the vulnerability of speech perception in young children with CIs whose everyday is comprised of listening in a complex three-dimensional noisy acoustic environment. “

  1. Understandably, the NH children show an overall higher response than CI children to auditory stimuli. For CI users, listening to speech in noise is too difficult to attend to the stimuli. Listening to music is not entirely different from listening to noise. And the result of the study shows the stimuli used in this study were not suitable for CI children to draw their attention.

The results could indeed be understood as showing that the musical stimuli, when presented in noise, were not attracting enough attention and hence an increased pupil response, similarly to speech. However, we know too little about music perception in children with CIs to draw such conclusion. Moreover, attention to speech is driven by very different cognitive processes compared to music, simply because speech ultimately conveys meaning whereas music does not, at least not in the same way.

We therefore believe that not much more can be said about music perception in children with CIs and did not change the paragraph on music perception in the discussion.

Minor concerns:

  • Table 1. PTA -> CI threshold 

Corrected.

  • Table 1, Age at CI2(y) -> (m) or convert the data as of year format

There was an inconsistency in expressing ages that we didn’t notice before; all ages are now expressed in months.

  • I do not think supplementary materials 1 and 2 are necessary.

We believe the appendix to be informative about the details of the presented analyses. We have therefore left the Appendix for now and we leave the decision to the Editor whereas to include it into the publication or not. If the appendix is not included, it can be added to the raw material and scripts at the OSF depository site.
